# Therapy of Subclinical Mastitis during Lactation

**DOI:** 10.3390/antibiotics11020209

**Published:** 2022-02-07

**Authors:** Scott McDougall, Laura M. Clausen, Hassan M. Hussein, Chris W. R. Compton

**Affiliations:** 1Cognosco, Anexa Veterinary Services, Morrinsville 3340, New Zealand; lmclausen@gmail.com (L.M.C.); h-hussein@xtra.co.nz (H.M.H.); 2School of Veterinary Science, Massey University, Palmerston North 4442, New Zealand; C.W.Compton@massey.ac.nz

**Keywords:** mastitis, subclinical, therapy, cloxacillin, duration, frequency, *Staphylococcus aureus*, *Streptococcus uberis*

## Abstract

This study tested the hypothesis that increasing the duration and/or frequency of antimicrobial treatment of subclinical mastitis would result in a higher bacteriological cure rate. Glands with a positive California mastitis test (CMT) from cows with an elevated somatic cell count (>500,000 cells/mL) that had an intramammary infection were randomly assigned at cow level to no treatment (Control; *n* = 80 glands), intramammary infusion of 200 mg cloxacillin sodium on three occasions at 48 h intervals (3 × 48 h; *n* = 273 glands), five occasions at 24 h intervals (5 × 24 h; *n* = 279 glands), or on five occasions at 48 h intervals (5 × 48 h; *n* = 72 glands). Glands were resampled at 21 (±3) and 28 (±3) days after initiation of treatment. The gland-level cure rate for any pathogen was 5/80 (6.2%), 139/173 (49.8%), 172/297 (61.6%) and 58/72 (80.6%) for Control, 3 × 48 h, 5 × 24 h and 5 × 48 h, respectively. The cure rate for major pathogens (defined as *Staphylococcus aureus* or *Streptococcus* spp.) was 4/52 (7.7%), 84/197 (42.6%), 96/183 (52.5%) and 36/48 (75%) for Control, 3 × 48 h, 5 × 24 h and 5 × 48 h, respectively. We conclude that treatment was superior to no treatment, and bacteriological cure rate was higher with the 5 × 24 h protocol than for the 3 × 48 h protocol and was higher with the 5 × 48 h than the 5 × 24 h protocol.

## 1. Introduction

Mastitis results in a range of clinical signs, from subclinical mastitis through to grossly evident local or systemic disease [1]. While clinical mastitis is commonly treated with antimicrobials and nonsteroidal anti-inflammatory drugs, subclinical cases are not commonly treated. However, there is production loss associated with subclinical infection [2], and these infections may act as a source of infection for other animals in the herd. Treatment of subclinical cases increases bacteriological cure rates [3,4], reduces SCC [3,5], reduces incidence of clinical mastitis [3,4] and potentially reduces forward transmission [6] compared with leaving quarters untreated.

Bacteriological curing of mastitis following antimicrobial therapy is affected by cow age [7], gland position (front vs. rear gland), the number of glands within a cow involved, the previous SCC, dry cow therapy treatment history [7,8,9], the pathogen [7,10], antimicrobial resistance [5,11,12] and presence of gland/teat end damage. 

For time-dependent antimicrobials such as beta-lactams, macrolides, and lincosamides, maintaining concentrations of the antimicrobial above MIC between 40–100% of the dosing interval and for a longer period are associated with higher bacteriological cure rates [13,14]. Bacteriological cure rates of subclinical intramammary infections were 10.5%, 38.8%, 53.7%, and 75.8%, respectively, following 0, 2, 5, or 8 days of intramammary treatment with 125 mg/day of ceftiofur [15]. Similarly, bacteriological cure rates of *Staph. aureus* of 6%, 56% and 86% were achieved following zero, two or eight intramammary treatments with 50 mg/day of pirlimycin, respectively [4], and cure rates of naturally acquired *Staph. aureus* or Streptococcus species infection were 0%, 44%, 61%, and 95% following treatment with nothing, or two, five, or eight daily intramammary doses of 50 mg of pirlimycin, respectively [16]. Similarly, the bacteriological cure rates were 16%, 32%, and 56% with no treatment, or following parenteral treatment with penethamate on three or six occasions at 24 h intervals, respectively, in subclinical glands [5]. However, a more recent study found no difference in bacterial cure rate between clinical mastitis cases assigned to three intramammary infusions at 12-hourly intervals compared with five infusions at 12-hourly intervals with amoxicillin clavulanic acid [17]. Similarly, similar bacteriological cure rates and similar survival rates were found following 3 × 12 hourly amoxicillin, 5 × 24 hourly amoxicillin or 5 × 24 hourly ceftiofur intramammary treatments of Gram-positive clinical mastitis cases [18].

Penicillin-resistant *Staph. aureus* is common amongst bovine mastitis isolates [19,20,21]. Resistance is conferred by beta-lactamases which are produced by the *blaZ* gene, which is commonly located on plasmids, but may also be located on chromosomes [22]. Cloxacillin is an anti-staphylococcal isoxazoyl penicillin that is beta-lactamase (pencillinase)-stable and hence may be effective against *Staph. aureus* strains producing this enzyme [23]. Cloxacillin has been extensively used as an intramammary therapy for mastitis, both during lactation [24,25] and at the end of lactation as dry cow therapy [26,27]. 

The primary objective of this study was to evaluate the bacteriological cure proportion and change in gland-level SCC following two durations and two frequencies of therapy with intramammary cloxacillin for treatment of naturally acquired subclinical mastitis cases on commercial dairy farms. It was hypothesized that treatment of subclinical mastitis with cloxacillin would result in a superior bacteriological cure rate than no treatment, and that increasing duration and/or frequency of treatment would result in higher bacteriological cure rates.

## 2. Results

### 2.1. Pretreatment Bacteriology

In total, 33% of all CMT-positive glands sampled were bacteriological-negative. Of the bacteriological-positive glands, the most prevalent pathogen isolated was *Staph. aureus* (*n* = 270, 37.7%), followed by *Corynebacterium spp.* and *Strep. uberis* (Table 1). A further 68 *Staph. aureus* isolates were recovered from mixed intramammary infections (IMI) from pretreatment samples. When gland results were grouped by pathogen types (major *= Staph. aureus, Strep. uberis, Strep. dysgalactiae, Strep. agalactiae*, *E. coli*, *Nocardia* spp.; minor = non-aureus staphylococci, Corynebacterium spp., Gram-negative rods) there was no difference in their distribution among treatment groups (*p* = 0.32).

Antimicrobial resistance was found in 128/338 (37.9%, 95% CI = 32.7–43.3%) of *Staph. Aureus* isolates for ampicillin and penicillin. Penicillin-resistant *Staph. aureus* isolates were found in 13 of the 38 (34.2%, exact 95% CI = 19.6–51.4%) herds enrolled, with the within-herd prevalence varying from 9 to 88%. No isolates were resistant to cephalothin, novobiocin, or oxacillin.

### 2.2. Bacteriological Cure

The crude proportion of glands with bacteriological cure of all pathogens or major pathogens only following no treatment (Control), 3 × 48 h, 5 × 24 h and 5 × 48 h interval cloxacillin treatments was 5/80 (6.3%) and 4/52 (7.7%), 136/273 (49.8%) and 90/204 (44.1%), 172/279 (61.6%) and 100/183 (54.6%), 58/72 (80.6%) and 37/48 (77.1%), respectively (Table 1).

In the final model, the bacteriological cure rate differed amongst each treatment group for all pathogens, and amongst major pathogen IMI (Figure 1a, Appendix A). Bacteriological cure rate for all pathogens, and for major pathogen infections, was lower for cows >3 years of age compared with those that were 2–3 years of age (Figure 1b). Bacteriological cure rate declined with increasing quarter-level SCC prior to treatment (Figure 1c).

Amongst *Staph. aureus* IMI, the bacteriological cure rate was 2.4 (95% CI = 0.0–7.3), 29.8 (95% CI = 22.0–37.7), 49.3 (95% CI = 40.2–58.4), and 79.8 (95% CI 65.4–94.2)% following no treatment (control), or 3 × 48 h, 5 × 24 h and 5 × 48 h interval cloxacillin treatments, respectively. The bacteriological cure rates differed between each treatment group (*p* < 0.05). Cure proportions were not affected by penicillin resistance, with 44/119 (37.0%) versus 77/195 (39.5%) cure rate for *Staph. aureus* IMI that were resistant and sensitive to penicillin, respectively. Bacteriological cure declined with increasing quarter level SCC prior to treatment (*p* = 0.005) and was higher in 2- and 3-year-old compared with >3-year-old animals (59.1 (95% CI 43.3–74.9) versus 36.5 (95% CI 31.3–41.8)% bacteriological cure rate for 2- and 3-year-olds versus >3-year-old cows, respectively; *p* = 0.02, Appendix A).

Amongst Streptococcal IMI, the bacteriological cure rates were 14.3 (95% CI = 0–32.8), 69.8 (95% CI = 56.8–82.8), 67.2 (95% CI = 54.3–80.0), and 61.6 (95% CI = 32.7–90.5)% following no treatment (control), or 3 × 48 h, 5 × 24 h and 5 × 48 h interval cloxacillin treatments, respectively. The bacteriological cure rates differed between the control group and each of the treatment groups (all *p* < 0.05), but the three cloxacillin treatment groups did not differ from each other (*p* > 0.05). No other explanatory variables were present in the final model (Appendix A).

### 2.3. New Infection Rate

The cumulative incidences of new IMI with any pathogen or a major pathogen between enrolment and the final Day 28 sample were 150/704 glands (21.3%) and 46/704 glands (6.5%), respectively. The cumulative incidence for any pathogen did not differ between treatment groups (*p* = 0.46), but the cumulative incidence of new major pathogens tended to be higher (*p* = 0.10) in Control glands (11/80, 13.8%) than in 3 × 48 h treatment glands (14/273, 5.1%). This is equal to overall incidence rates of 7.6 and 2.3 new IMI for any pathogen or major pathogen-only IMI per gland per 1000 days at risk.

### 2.4. Quarter-Level Somatic Cell Count

The median gland somatic cell count did not differ between groups on the day of enrolment but was lower for treatment groups 5 × 24 h and 5 × 48 h compared to control glands (*p* < 0.01 for each comparison) at Days 21 and 28 (*p* ≤ 0.01 for each comparison) with the 3 × 48 h group intermediate (Figure 2a). The median gland somatic cell count was lesser for glands that were cured of any pathogen compared to those that were not on each day of sampling (*p* < 0.01 for each comparison; Figure 2b).

### 2.5. Incidence of Clinical Mastitis within 28 Days of Enrolment

The incidence of clinical mastitis within 28 days of enrolment did not differ between treatment groups (0/81 (0%), 9/298 (3.0%), 9/292 (3.1%), and 1/77 (1.3%) for quarters with subclinical mastitis following no treatment (Control), 3 × 48 h, 5 × 24 h and 5 × 48 h interval cloxacillin treatments, respectively; *p* = 0.36).

## 3. Discussion

This negative and positive controlled, multiherd intervention study of treatment of subclinical mastitis by intramammary infusion of cloxacillin demonstrated that increasing either the frequency and/or duration of therapy resulted in increased bacteriological cure rate. 

Sufficient cases were enrolled to test the hypotheses, treatment groups did not differ in terms of age, breed, teat end score, pathogen, and cows from a large number of herds were enrolled. Thus, the external validity of the study is high, and there was no bias in group allocation.

### 3.1. Effect of Duration and Frequency of Treatment on Cure Rate

The increasing cure rate observed with increasing duration and/or frequency of treatment agrees with a growing body of data showing that increasing the duration of treatment increases cure rate for the time-dependent antibiotics [4,5,15]. Over all pathogens, major pathogens, and *Staph. aureus* IMI, the 5 × 48 h treatment group resulted in higher bacteriological cure than the 5 × 24 h treatment group. Like other beta lactams, cloxacillin is a time-dependent antimicrobial [13,14], and as the concentrations of cloxacillin in cisternal milk are likely to being above the MIC for most mastitis pathogens for up to 48 h, the 5 × 48 h treatment protocols likely resulted in concentrations above MIC for a significantly longer period than following the 5 × 24 h treatment with a resultant increase in bacteriological cure rate.

Similarly, over all pathogens, major pathogens, and *Staph. aureus* IMI, the 5 × 24 h treatment group resulted in higher bacteriological cure than the 3 × 48 h treatment group. While the timing of these treatments resulted in the duration over which treatments were applied being similar (i.e., the final treatment occurred 96 h after the first treatment irrespective of whether three or five doses were given), it is possible that higher concentrations were achieved with the 5 × 24 h dosing frequency, and associated with this, the total duration over which the cloxacillin concentration was greater than MIC may have been longer. Following infusion of a “slow release” formulation of cloxacillin, concentrations had not returned to basal level by 48 h post-intramammary-infusion [24]. Thus, where dosing was occurring at 24-hourly intervals, it is likely that the maximum concentrations achieved were higher than in the 3 × 48 h treatment group. Additionally, due to a diffusion gradient within the mammary gland following intracisternal infusion of antimicrobials, and the limited distribution of cloxacillin within the mammary gland, it is possible that the higher concentrations potentially achieved by the higher frequency dosing (i.e., 24 versus 48 hourly) may have resulted in higher concentrations at the dorsum of the udder with potential increases in the bacteriological cure rate. 

Amongst glands infected with Streptococcus (that is, *Strep. agalactiae*, *Strep. dysgalactiae* or *Strep. uberis*), all cloxacillin treatment groups resulted in a higher bacteriological cure than for the control group. However, as distinct from *Staph. aureus*, there was no difference in bacteriological cure rate between the 3 × 48 h, 5 × 24 h, and 5 × 48 h groups for *Streptococcus spp.* infections. Similarly, treatment of naturally acquired environmental streptococcal infection with zero, two, five, or eight daily doses of ceftiofur resulted in the 5- and 8-day treatments having higher bacteriological cure rates than the control group, but the cure rates amongst the three treatment groups did not differ [15]. Conversely, bacteriological cure rates of *Strep. uberis* infections increased following zero, two, or eight daily pirlimycin treatments [4]. Care must be taken in the interpretation of these data due to the relatively small sample size (i.e., only 41, 92, and 132 glands were infected with Streptococcus in [4,15], and the current study, respectively). It is not clear why no increase in cure rate occurred with the longer duration treatment protocols in the current study for Streptococci. Another study [24] reported no difference in bacteriological cure rate of Streptococci following comparison of two doses (0.2–0.6 g), and one versus three treatments at 48 h intervals, or with two formulations of cloxacillin. However, this was partly due to the very high (>90%) bacteriological cure rates observed for *Strep. agalactiae* and *Strep. dysgalactiae* irrespective of dose rate, frequency, or formulation, suggesting that even short (single)-duration treatment of 0.2 g of “quick release” formulation were highly effective against these Streptococci. The bacteriological cure rate for *Strep. uberis* overall in that same study was 82%, with little variation associated with dose rate, number of doses or release formulation. The MIC_90_ of *Strep. dysgalactiae* and *Strep. uberis* isolated from cows diagnosed with mastitis in New Zealand were 0.25 and 4.0 µg/mL, respectively, [21]. Similarly, MICs for *Strep. uberis* for cloxacillin of 4 µg/mL were reported from France [28] and an MIC of 2 µg/mL for oxacillin (the preferred class antimicrobial) from Germany [29]. Mutations in penicillin binding protein 2× are associated with an elevated cloxacillin MIC and poorer bacteriological cure rates amongst *Strep. uberis* isolates [30,31]. Concentrations of cloxacillin in milk following infusion of 200 mg of a “slow release” formulation on three occasions at 48 h intervals peaked at approximately 30 µg/mL, and remained above 4 µg/mL for 7 days [24], suggesting that concentrations of cloxacillin achieved following intramammary infusion were likely above the *Strep. uberis* MIC for an extended period. As neither the MIC of the *Strep. uberis* isolates nor the concentrations cloxacillin in milk were determined in the current study, it is not clear whether some *Strep. uberis* isolates have MICs such that cloxacillin concentrations achieved are not optimal.

### 3.2. Effect of Age and SCC Duration on Cure Rate

The cure rate declined with age of the cow and with increasing SCC prior to treatment. Declining cure rates have been associated with increasing cow age following treatment of both subclinical [4,32] and clinical mastitis [7,33]. However, another study of clinical mastitis treatment did not find an association between age and cure rate [34]. Age may be a proxy for chronicity of infection, which may affect bacteriological cure rate. In the current study, effects of age were present in the models considering all intramammary infections and major pathogen infections, but not in the model of Streptococci. This may reflect differences in the epidemiology of Streptococci and *Staph. aureus*, as Streptococci behave predominantly as environmental pathogens, meaning prevalence of streptococcal infection may not vary with age, but in contrast, *Staph. aureus,* due to its contagious nature, may be more prevalent and of greater duration in older animals, with increasing chronicity associated with lower cure rates.

The cure rate declined with increasing gland-level SCC before treatment in the current study, as has been demonstrated in previous studies [32,34]. The higher SCC may be associated with more severe infections, potentially related to the virulence of the pathogen.

Stage of lactation (days in milk) was not associated with cure in the current study. This contrasts with several other studies that have demonstrated effects of stage of lactation on cure. There was a stage of lactation by treatment interaction in a study assessing zero, two and eight daily treatments with pirlimycin, whereby cure rate declined with stage of lactation in untreated controls, whereas there was an increased cure rate with increasing stage of lactation amongst animals treated for 2 days with pirlimycin, but this effects occurred only in the first 100 days of lactation (relative to >200 days) amongst those cows treated for 8 days [4]. Similarly, bacteriological cure rate was lower in the first 100 days of lactation compared to in cows >200 days of lactation amongst subclinical *Staph. aureus* cases [32]. In the current study, most cows (80%) were <100 days in milk and the median days in milk was 69 days. Thus, there is limited power in the current study to demonstrate any effect of stage of lactation, if present.

### 3.3. Effect of Penicillin Resistance amongst Staph. aureus Isolates on Cure Rate

Penicillin resistance was present in 38% of the *Staph. aureus* isolates tested in the current study. Previous New Zealand studies have reported penicillin resistance of bovine mastitis *Staph. aureus* isolates to be in the range of 25% and 38% [21,35,36]. Presence of penicillin resistance did not affect cure rate of *Staph. aureus* in the current study. The sample size was relatively small for testing this hypothesis with post hoc analysis, finding that with 120 cases/group, a difference of 22% versus 37% in bacteriological cure rate could have been detected. However, as the antibiotic used in the current study (cloxacillin) is stable against beta-lactamases (a common mechanism of resistance in *Staph. aureus*), this result is not unexpected. Conversely, where glands infected with *Staph. aureus* were treated with a narrow-spectrum penicillin which was susceptible to beta-lactamases, the bacteriological cure rates in resistant isolates were significantly lower than those that were sensitive [5]. However, other studies have shown that penicillin resistance has been associated with poorer bacteriological cure rates [7,32,37], irrespective of the antimicrobial used. In those studies, as in the current one, the antibiotics used were stable to beta-lactamases, and hence the effect of decreased cure rate in the beta lactamase producing isolates was independent of the antimicrobial used. It has been hypothesized that bacteria that have the beta-lactamase genes may also carry other virulence factors and hence it is these other factors, rather than the presence of the beta lactamase per se, that accounts for the lower cure rates [38].

### 3.4. Effect of Treatment on Gland-Level SCC

The SCC at gland level were lower in glands treated with cloxacillin at 5 × 48 and 5 × 24 h intervals, than the control glands. Bacteriological cure resulted in lower gland-level SCC. This agrees with previous studies demonstrating that extension of duration of therapy is associated with increased pathological cure rate, and hence lower SCC [3,4]. It should be noted that <20% of glands had SCC of <200,000 cells/mL by Day 28 post-treatment. In practical terms, cows treated for subclinical mastitis may still have elevated SCC at the time of return to milk supply, and where several cows in a herd have been treated, this may increase the bulk milk SCC.

## 4. Materials and Methods

This prospective, randomized, positive- and negative-controlled study was undertaken using cows from 44 spring-calving, predominantly pasture-fed dairy cows from the Waikato region of New Zealand. Cows were milked twice a day. The herds were a convenience sample from amongst herds serviced by one veterinary business (Anexa) that was willing to follow the study protocol, to undertake at least 3 production recordings per lactation, and that was allowed access to electronic and hard-copy cow records and signed an informed consent form.

### 4.1. Cow Procedures

Cows were selected based on having a cow-composite somatic cell count (SCC) of >500,000 cells/mL at a production (herd-test) recording within the preceding 14 days. Cows were excluded if they had been treated with antimicrobials in the preceding 30 days. Remaining cows were separated from the herd between milkings and examined by experienced technicians. Cows with one or more teat ends with very severe hyperkeratosis [39] or which had one or more nonfunctional or “light” quarters, that is, glands visually estimated to have <80% of the volume of other glands within the cow, or did not have at least one gland positive (i.e., greater than trace) on the Californian Mastitis test (CMT) were excluded. A cow could only be enrolled once in the study, and cow enrolment occurred between September 2010 and December 2011.

Glands that were CMT-positive from amongst the cows meeting these enrolment criteria had duplicate foremilk milk samples collected for subsequent microbiology and quarter-level SCC determination. Samples for microbiology were held at −20 °C before processing, samples for SCC determination were preserved with one drop of bronopol, and the samples were held at 4 °C for up to 7 days before submission to the laboratory.

Cows with at least one gland meeting the inclusion criteria were randomly assigned to the negative control group (i.e., no treatment), 3 intramammary infusions at 24 h intervals (3 × 24 h), 5 treatments at 24 h interval (5 × 24 h), and 5 treatments at 48 h intervals (5 × 48 h) in a 1:4:4:1 pattern within blocks of 10 cows within each herd. Where more than one gland within cow met the inclusion criteria, all glands within the cow were treated the same way. The treatment was one of two commercial batches of a 200 mg cloxacillin sodium formulation (Orbenin LA, Zoetis Animal Health, Auckland, New Zealand Animal Compounds Veterinary Medicine registration number A3664). Following aseptic teat-end preparation by scrubbing with a cotton wool ball soaked in 70% methylated spirits, treatment was administered using the “partial insertion” method, that is, the tip of the treatment cannula was inserted approximately 3 mm into the teat canal, the end of the teat canal was included with a thumb and forefinger and the antimicrobial gently infused. The antimicrobial was manually dispersed within the mammary gland by massage. Following infusion, teat antiseptic (0.25% available iodine) was applied. The milk withholding period was 7 or 8 milkings following the 3 × 48 h treatment, and the 5 × 24 h and 5 × 48 h, respectively.

All enrolled glands were re-examined by trained technicians 21 (±3) and 28 (±3) days after the first treatment. A single milk sample (~5 mL) was collected for bacteriology and a further sample (~25 mL) collected for gland-level SCC determination. Additionally, the presence or absence of changes to the milk (i.e., flecks or clots or wateriness of the milk) and/or heat/swelling of the mammary gland were recorded at these time points (Figure 3).

Herd owners and staff were asked to monitor for presence of clinical mastitis (i.e., presence of changes to the milk such as presence of flecks or clots) and/or changes to the mammary gland (heat/swelling) from the time of initiation of treatment through to Day 28. Where clinical mastitis was detected, the herd owner treated such quarters with antimicrobials and recorded the cow, gland position, and treatment, and these data were later recovered for analysis.

A total of 1171 cows were considered eligible for the study based on SCC data, of which 518 were excluded, 276 related to recent antimicrobial usage, 93 due to presence of light or nonfunctional glands, 69 because none of the glands were CMT positive, 47 due to presence of very rough teat-end hyperkeratosis and 33 as they were systemically ill or had only recently calved. Hence, 653 cows were randomly allocated to the treatment groups. A further 14, 90, 87, and 15 cows in the Control, 3 × 48 h, 5 × 24 h, and 5 × 48 h groups, respectively, were excluded due to no bacterial growth of the selected quarter, or the samples being defined as contaminated or culturing a yeast in the laboratory.

Data from 704 glands from 447 cows from 37 herds were considered for analysis. The distribution of cows among the treatment groups was 50 (11%) in the control group, 173 (39%) in the 3 × 48 h group, 175 (39%) in the 5 × 24 h group and 49 (11%) in the 5 × 48 h group (Appendix A). The distribution of cow age and breed categories did not differ between treatment groups (*p* = 0.55 and 0.63, respectively). The median days in milk at enrolment was not different between treatment groups (*p* = 0.84), but the median SCC (at cow level) at the most recent herd test prior to enrolment differed between treatment groups (*p* = 0.006; Appendix A). 

At gland level, 80 (11%) glands were in the control group, 273 (39%) in the 3 × 48 h group, 279 (40%) in the 5 × 24 h group and 72 (10%) in the 5 × 48 h group. The distribution of gland position (*p* = 0.95), teat-end score (*p* = 0.59), and CMT score (*p* = 0.58) at enrolment did not differ among the treatment groups (Appendix A). The median SCC at individual gland level did not differ between treatment groups (5501 (345–27,109), 4730 (43–30,570), 4209 (49–29,758), and 4393 (219–22,299) median (minimum–maximum) SCC (×1000 cells/mL) for quarters assigned to be in the Control group, 3 × 48 h, 5 × 24 h and 5 × 48 h, respectively *p* = 0.57). 

This study was blinded by not including the treatment group on lists of animals in glands to be sought to be sampled following the initial treatment, and all samples were identified with unique sample identification that did not include treatment group.

### 4.2. Laboratory Techniques

#### 4.2.1. Microbiology

Microbiology was undertaken following the procedures recommended by the National Mastitis Council, USA [40]. Briefly, 10 μL of milk was streaked onto a quarter of a 5% blood agar plate containing 0.1% esculin (Fort Richard, Auckland, New Zealand), and incubated at 37 °C for 48 h. The genus of bacteria was determined based on colony morphology, Gram stain, esculin reaction, coagulase, and CAMP tests. Coliforms were sub-cultured on MacConkeys agar, and an oxidase test performed. 

Non-aureus staphylococci (NAS) and *Corynebacterium* spp. were defined as ‘minor’ pathogens while *Staph. aureus*, *Streptococcus agalactiae*, *Streptococcus dysgalactiae*, *Strep. uberis, Nocardia spp.* and *Escherichia coli* were defined as ‘major’ pathogens. 

*Staphylococcus aureus* isolated from the pretreatment samples were kept on Dorset egg slopes (Fort Richard, Auckland, New Zealand) at 4 °C and antimicrobial sensitivity determined by zone diffusion testing using discs for ampicillin, cephalothin, novobiocin, oxacillin and penicillin following CLSI methodology [41].

#### 4.2.2. Somatic Cell-Count Determination

Somatic cell counting for individual gland samples and from composite milk samples from herd testing was performed using flouro-optic methods (Fossomatic 5000, Foss, Denmark) by the LIC Testlink Laboratory, Riverlea Rd, Hamilton, New Zealand.

### 4.3. Data Handling and Analysis

Data were stored in a custom-built relational database (Microsoft Access 2007, Redmond, WA, USA). Statistical analysis was performed using R [42] or STATA v17 (Stata Corp., College Station, TX, USA). Statistical significance was declared at *p* < 0.05.

The primary outcome variable was bacteriological cure of glands. This was defined as the presence of the bacterial species present prior to treatment (i.e., Day 0) at neither Day 21 nor 28 post-treatment. Where one or more microbiology results were defined as contaminated (i.e., >2 morphologically distinct bacterial colonies present) the bacteriological cure status of the gland could not be determined and was coded as “null”. A new intramammary infection was defined as occurring where a bacterial species was isolated at either Day 21 or 28 that was not present at Day 0. Note that a gland could be defined both as a cured and as having a new intramammary infection. 

Tables were made of descriptive statistics of the cows and glands in the final data set. Differences were examined by chi-squared test between cows in different treatment groups for categories of age (coded as 2–3 vs. >3 years of age), breed (coded as pure breed if ≥12/16th of that breed, otherwise crossbred), number of days in milk at enrolment and cow-level SCC prior to enrolment; and differences between glands enrolled in treatment groups for gland position (coded as fore vs. rear glands), teat-end hyperkeratosis score (i.e., none, smooth, or rough), and gland SCC (coded as <1 million, 1–4.9 million, 5–9.9 million, and >10 million cells/mL) at enrolment. Pairwise comparisons were adjusted for multiple comparisons using the Bonferroni correction. Somatic cell count was categorized for analysis, or analysis was performed by nonparametric tests because log-transformation did not adequately normalize the distribution.

Initially, univariate associations were tested by chi-squared test between the outcome variables (IMI cure proportion for all pathogens, IMI cure proportion for major pathogens, IMI cure proportion for *Staph. aureus*, and IMI cure proportion for Streptococci (i.e., *Strep agalactiae*, *Strep. dysgalactiae*, and *Strep. uberis* combined)) and possible predictor variables (treatment group, categories of age, breed, days in milk at enrolment, enrolment teat-end score, gland position, enrolment gland SCC and presence of penicillin resistance (for *Staph. aureus* isolates only)). Because the outcome variables could not be considered independent of each other and were measured at the gland-level, which were clustered within cows, accounting for this multilevel data structure was necessary in the final model to estimate treatment effects. Hence, generalized linear mixed models [43] for the outcome variables were built by manual forward stepwise addition of variables significant (*p* < 0.2) from the univariate analysis and known potential confounding variables, e.g., age category and days in milk at start of treatment. Treatment group was forced into the models as a fixed effect. Variables were retained in the final model where they were significant (*p* < 0.05) or were effect modifiers of the association between treatment and cure proportion (changed regression coefficient >15%). The probability of cure was predicted from the final models with adjustment for the effect of other retained variables and these results were then tabulated.

### 4.4. Power Analyses

Based on expected bacteriological cure rates in the negative control group of 20% and in the 5 × 48 treatment group of 80%, respectively [5], a priori 100, 400, 400, and 100 glands eligible for treatment within each of the control, 3 × 48 h, 5 × 24 h, and 5 × 48 h treatment groups were thought to be required, respectively.

## Figures and Tables

**Figure 1 antibiotics-11-00209-f001:**
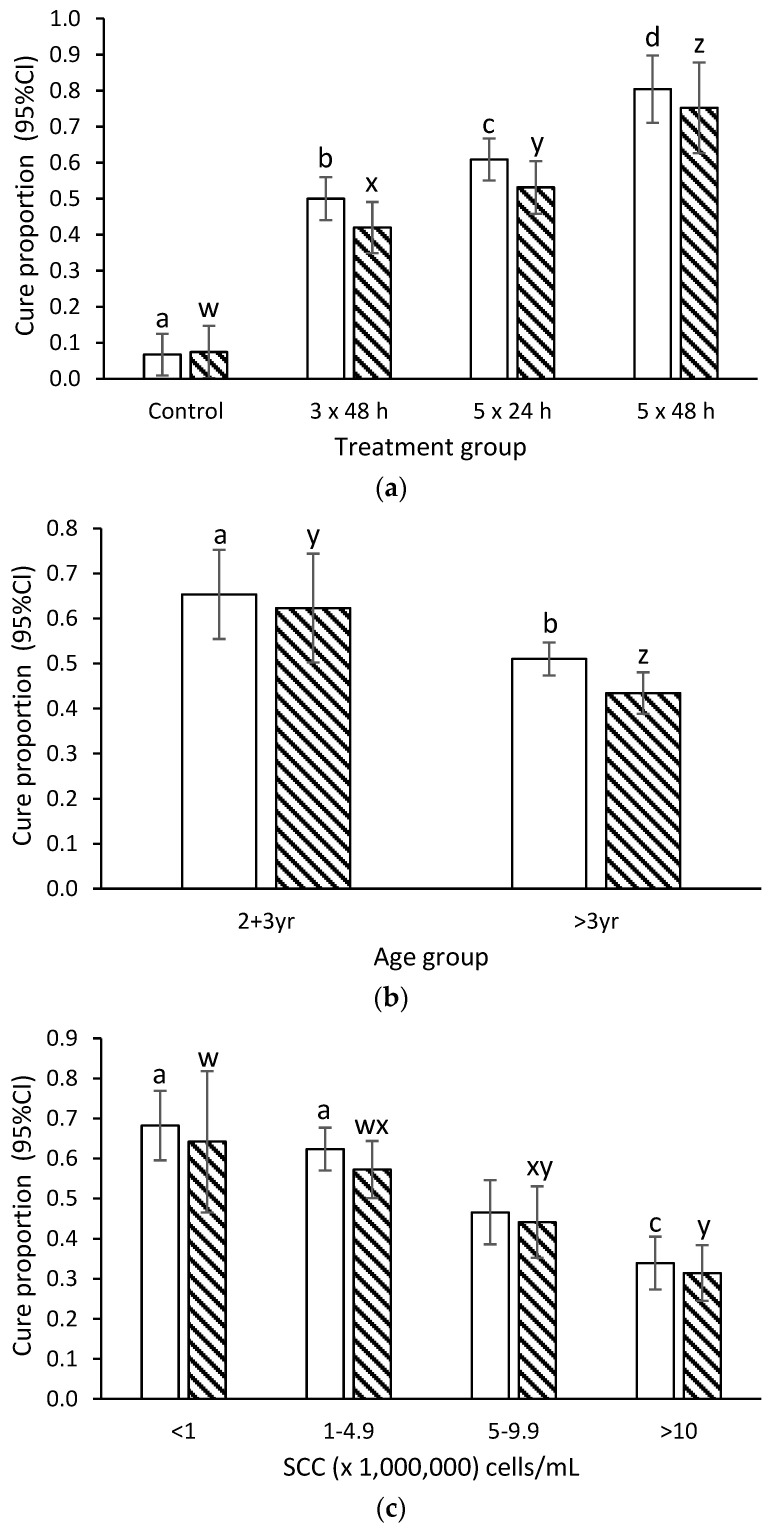
Estimated mean cure proportion (with 95% confidence limits) for treatment of subclinical mastitis with intramammary cloxacillin by (**a**) treatment group, (**b**) age, and (**c**) quarter-level SCC prior to treatment for any pathogen (open bars) or for major pathogens (cross hatched bars). Within pathogen type, columns with differing superscripts differ at *p* < 0.05.

**Figure 2 antibiotics-11-00209-f002:**
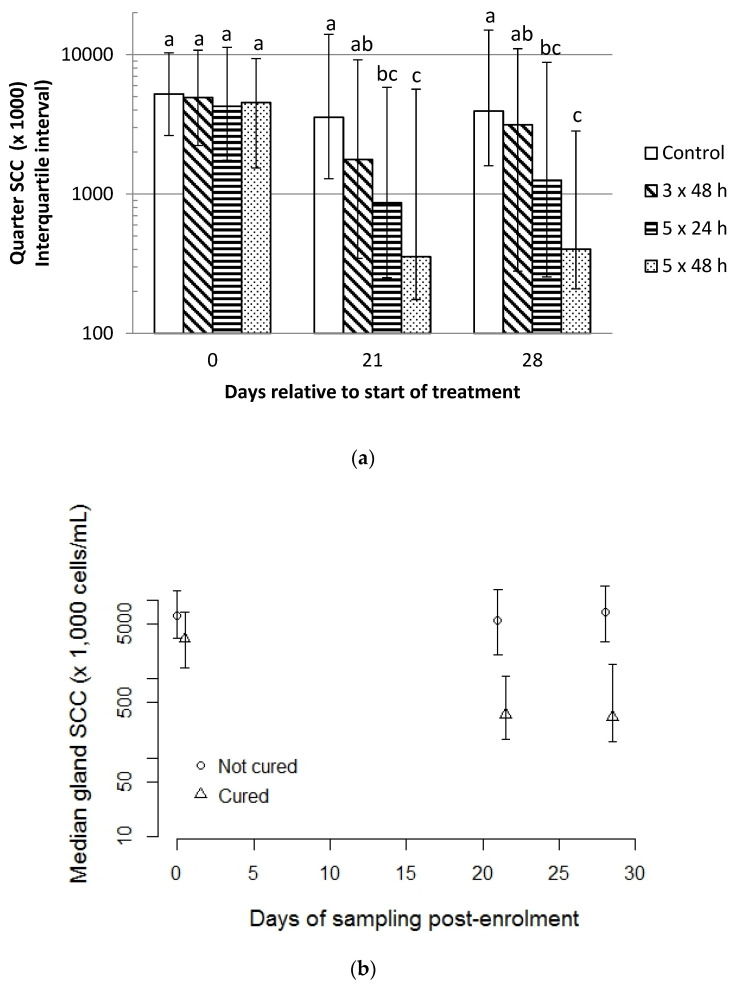
Median (interquartile range) of somatic cell count (SCC; ×1000 cells/mL) by (**a**) day post-treatment and by treatment group irrespective of bacteriological cure outcome, and (**b**) in glands that did or did not undergo bacteriological cure. Columns within days with superscripts differing differ at *p* < 0.05.

**Figure 3 antibiotics-11-00209-f003:**
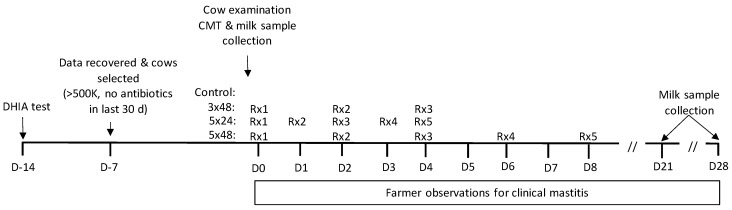
Diagram of cow procedures including data collation from dairy herd improvement testing (DHIA test), cow examination, California mastitis test (CMT) and treatment regime. Cows were allocated to no treatment (Control), 3 intramammary infusions of 200 mg cloxacillin at a 48 h interval (3 × 48), 5 treatments at 24 h interval (5 × 24), or 5 treatments at 48 hourly intervals (5 × 48). Rx indicates intramammary treatment.

**Table 1 antibiotics-11-00209-t001:** Distribution (number of glands (No.)) of bacteriology results before treatment, and bacteriological cure rate (%) by treatment group among bacteriological-positive glands treated with intramammary cloxacillin on 3 occasions at 48 h intervals (3 × 48 h), 5 times at 24 h intervals (5 × 24 h), or 5 times at 48 h intervals (5 × 48 h) or left as untreated controls following diagnosis with subclinical mastitis. The total number of glands and total number of cows are the number of glands and cows included in the final analyses.

		Control		3 × 48 h		5 × 24 h		5 × 48 h		Total	
		No.	% Cured	No.	% Cured	No.	% Cured	No.	% Cured	No.	% Cured
Minor	NAS ^1^	13	0.0	33	78.8	37	83.8	15	86.7	98	71.4
	*Corynebacterium* spp.	15	6.7	36	58.3	59	76.3	9	100.0	119	63.9
	All minor	28	3.6	69	68.1	96	79.2	24	91.7	217	67.3
Major	*Staph. aureus*	26	0.0	115	33.0	98	41.8	26	80.8	265	37.7
	*Strep. agalactiae*	1	100.0	1	100.0					2	100.0
	*Strep. dysgalactiae*	6	16.7	15	93.3	11	90.9	1	100.0	33	78.8
	*Strep. uberis*	8	0.0	36	61.1	42	59.5	11	54.5	97	54.6
	All *Streptococci*	15	13.3	52	71.2	53	66.0	12	58.3	132	61.4
	*E. coli*			2	100.0	2	100.0			4	100.0
	Mixed ^2^	11	18.2	33	33.3	26	69.2	9	88.9	79	49.4
	Gram -ve rods			1	100.0					1	100.0
	*Nocardia* spp.			1	100.0	4	100.0	1	100.0	6	100.0
	All majors	52	7.7	204	44.1	183	54.6	48	77.1	480	46.7
Total glands		80	6.3	273	49.8	279	61.6	72	80.6	704	52.7
Total cows		50		173		175		49			

^1^ Non-aureus staphylococcus (i.e., coagulase-negative staphylococcus), ^2, i.e.^, 2 distinct bacterial species isolated.

## Data Availability

Data is available upon request.

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
