# Peer review of "Therapy of Subclinical Mastitis during Lactation"

_antibiotics, 2022, doi:10.3390/antibiotics11020209_

Round 1
Reviewer 1 Report
The article is well written, sounds. I have only minor corrections which would improve the manuscript.
Minor corrections:
Authors stated that 447 cows were enrolled in the study. However the summarized value of cows of different age and different breed (Table 1) is equal to 445. Authors are asked either to explain the difference or to correct the table content.
Authors stated (line 533/534):”Tables were made of descriptive statistics of AMR test results, cumulative incidence of recurrence of clinical mastitis (clinical failure) and new infection rates.“ Data on AMR is not presented in table. Authors are asked to revise the sentence or to present data regarding the AMR in a new table.
Authors are asked to state the time of sampling (before ordinal milking or between milking interval) and the milk fraction used for somatic cell count determination and microbiology (foremilk or other fraction).
Also, I suggest that authors add several recent references in the Introduction section in order to improve it. References which I suggest in the manuscript are following:
Isolation of bovine clinical mastitis bacterial pathogens and their antimicrobial susceptibility in the Zenica region in 2017. Vet. stn. 51, 47-52. (In Croatian). doi.org/10.46419/vs.51.1.5
Bovine mastitis: a persistent and evolving problem requiring novel approaches for its control - a review. Vet. arhiv 88, 535-557. doi.org/10.24099/vet.arhiv.0116
Effect of feed additive supplementation on bovine subclinical mastitis. Vet. stn. 52, 445-460. doi.org/10.46419/vs.52.4.12
Investigation of the presence of slime production, VanA gene and antiseptic resistance genes in Staphylococci isolated from bovine mastitis in Algeria. Vet. stn. 52, 57-63. doi.org/10.46419/vs.52.1.9
Multi Locus Sequence Typing and spa Typing of Staphylococcus aureus Isolated from the Milk of Cows with Subclinical Mastitis in Croatia. Microorganisms 9, 725. doi: 10.3390/microorganisms9040725
Alternative treatment of bovine mastitis. Vet. stn. 52, 639-649. doi.org/10.46419/vs.52.6.9
Use of somatic cell count in the diagnosis of mastitis and its impacts on milk quality. Vet. stn. 52, 751-764. (In Croatian). doi.org/10.46419/vs.52.6.11
Author Response
Authors stated that 447 cows were enrolled in the study. However the summarized value of cows of different age and different breed (Table 1) is equal to 445. Authors are asked either to explain the difference or to correct the table content.
AU: the age and breed of 2 animals was not available and this has been noted in the Table 1 caption.
Authors stated (line 533/534):”Tables were made of descriptive statistics of AMR test results, cumulative incidence of recurrence of clinical mastitis (clinical failure) and new infection rates.“ Data on AMR is not presented in table. Authors are asked to revise the sentence or to present data regarding the AMR in a new table.
AU: reference to AMR test results has been removed.
Authors are asked to state the time of sampling (before ordinal milking or between milking interval) and the milk fraction used for somatic cell count determination and microbiology (foremilk or other fraction).
AU: Additional text has been added at line 426 to clarify that cows were drafted (separated) from the main mobs and examined between milkings. Additionally the fact that the milk samples were “foremilk” samples is now specifically stated at line 435.
Also, I suggest that authors add several recent references in the Introduction section in order to improve it. References which I suggest in the manuscript are following:
Isolation of bovine clinical mastitis bacterial pathogens and their antimicrobial susceptibility in the Zenica region in 2017. Vet. stn. 51, 47-52. (In Croatian). doi.org/10.46419/vs.51.1.5
Bovine mastitis: a persistent and evolving problem requiring novel approaches for its control - a review. Vet. arhiv 88, 535-557. doi.org/10.24099/vet.arhiv.0116
Effect of feed additive supplementation on bovine subclinical mastitis. Vet. stn. 52, 445-460. doi.org/10.46419/vs.52.4.12
Investigation of the presence of slime production, VanA gene and antiseptic resistance genes in Staphylococci isolated from bovine mastitis in Algeria. Vet. stn. 52, 57-63. doi.org/10.46419/vs.52.1.9
Multi Locus Sequence Typing and spa Typing of Staphylococcus aureus Isolated from the Milk of Cows with Subclinical Mastitis in Croatia. Microorganisms 9, 725. doi: 10.3390/microorganisms9040725
Alternative treatment of bovine mastitis. Vet. stn. 52, 639-649. doi.org/10.46419/vs.52.6.9
Use of somatic cell count in the diagnosis of mastitis and its impacts on milk quality. Vet. stn. 52, 751-764. (In Croatian). doi.org/10.46419/vs.52.6.11
AU: Thank you for those suggestions. We have not included them as the other referee asked us to reduce the manuscript length.
Reviewer 2 Report
Introduction
The first paragraph can be deleted, as it includes basic knowledge.
Lines 52-68 can be usefully shortened.
The hypothesis of the authors should be clearly presented.
Materials and methods
4.1. Please presented inclusion and exclusion criteria clearly at the start of the sub-section.
Please add a figure with a timeline of the study.
4.3. For each multivariable model, please present tables in supplementary material with number of variables included in each one and a list of variables that were included in the final model.
Results
Figures. Please be considerate to colour-blind readers. Please use a monochromatic palette with shades and motifs for noting groups etc. All figures should be redone taking into account that some readers will not be able to understand them when using colours.
Sub-section 2.1. can be part of Ma & Me.
Tables 3 and 4 are very detailed and can be transferred to supplementary material.
2.3 Please include a table with summary of the results.
Discussion
The discussion is very long and verbose and this makes it difficult for evaluation.
Please rewrite, by reducing in length and also adding sub-sections to make it reader-friendly. Also, please avoid locally used expressions that cannot be understood by an international readership.
The manuscript should be reevaluated after revision.
Author Response
Introduction
The first paragraph can be deleted, as it includes basic knowledge.
AU: The paragraph has been edited
Lines 52-68 can be usefully shortened.
AU: the last 3 sentences of that paragraph have been removed
The hypothesis of the authors should be clearly presented.
AU: It is not clear what the reviewer is asking here as lines 62-65 state that increasing either the duration and/or the frequency of treatment would increase bacteriological cure rate.
Materials and methods
4.1. Please presented inclusion and exclusion criteria clearly at the start of the sub-section.
AU: The first paragraph in this section is specifically dealing with inclusion/exclusion. The process was multistep as there was a pre-screen based on existing DHI data (i.e. SCC and treatment records) which resulted in exclusion of some animals immediately, followed by the cow level and quarter level examinations to exclude those with teat end hyperkeratosis, light glands et cetera. This paragraph is written in such a way to make it transparent that this two-step process was used.
Please add a figure with a timeline of the study.
AU: A new figure (Figure 3) has been added.
4.3. For each multivariable model, please present tables in supplementary material with number of variables included in each one and a list of variables that were included in the final model.
AU: The list variables offered for each model is included in the text in section 4.3 (lines 514-531). The variables included in the final model had been included in the text, but 4 supplementary tables have now been added providing output of the modelling of cure rate for any pathogen, for major pathogens only, for Staphylococcus aureus only, and for the streptococci.
Results
Figures. Please be considerate to colour-blind readers. Please use a monochromatic palette with shades and motifs for noting groups etc. All figures should be redone taking into account that some readers will not be able to understand them when using colours.
AU: Done
Sub-section 2.1. can be part of Ma & Me.
AU: Section 2.1 has been incorporated in the materials and methods
Tables 3 and 4 are very detailed and can be transferred to supplementary material.
AU: Done
2.3 Please include a table with summary of the results.
AU: the raw cure rate data is included in what is now called Table 1
Discussion
The discussion is very long and verbose and this makes it difficult for evaluation.
Please rewrite, by reducing in length and also adding sub-sections to make it reader-friendly. Also, please avoid locally used expressions that cannot be understood by an international readership.
AU: The discussion has been edited. As requested subheadings have been added to improve readability
Round 2
Reviewer 2 Report
The manuscript has been duly improved after revision of the initial version.
The authors can consider to include some recently published relevant references, which can fit within the remit of this manuscript.
After this improvement, the manuscript can be accepted for publication.
Author Response
Hi
The suggested formatting and other changes have been accepted in the attached revised manuscript.
Additionally two more recent studies that have examined duration of therapy on back to all cure rate of mastitis cases. have been added to the introduction (in yellow) (McDougall et al. 2009; Tomazi et al. 2021).
Yours
Scott McDougall